# Food insecurity and water management shocks in Saudi Arabia: Bayesian VAR analysis

Raga M. Elzaki[1,2]*, Mohammed Al-Mahish[1]

1 Department of Agribusiness and Consumer Science, College of Agriculture and Food Science, King Faisal University, Al Ahsa, Saudi Arabia, 2 Department of Rural Economics and Development, Faculty of Animal Production, University of Gezira, Wad Madani, Sudan

* rmali@kfu.edu.sa

**Data Availability Statement:** All relevant data are within the manuscript and its Supporting Information files.

**Funding:** This research was funded by the Deanship of Scientific Research, King Faisal

## Abstract

The existing conditions of domestic agricultural production and the resulting products will not be able to fruitfully address the increasing food demand due to the limited fertile land and water resources in Saudi Arabia. Moreover, the escalating threat of a hotter climate, the deterioration in precipitation, and harsh droughts in Saudi Arabia have reduced the predictability of water management efficiency and resulted in the exhaustion of water bodies and serious degradation of ecosystems that have directly affected agricultural systems and indirectly, food security. This study also aims to assess the impact of water efficiency on food insecurity in Saudi Arabia. The study applied the Bayesian Vector Autoregressive (BVAR) model for the reference period for the data extended from 2000–2020. Likewise, we used both impulse response functions (IRFs) and forecasting variance error decomposition (FVED) through 1000 Monte Carlo simulations according to the BVAR model to examine both the response of food insecurity to the shocks on water management efficiency used for various purposes and the decomposition of error variance in food insecurity. The results show that food insecurity was not observed throughout this study. The results of the BVAR analysis indicate that in the short run, the coefficients of water use efficiency are significant based on the Food Insecurity Multidimensional Index (FIMI). Also, the BVAR model provides a better forecast with an interdependence on water use efficiency for agricultural purposes and FIMI. Moreover, the results obtained from IRFs have shown a significant effect of water efficiency on FIMI. Water use efficiency for agriculture and industrial purposes reduces food insecurity while increasing water for services use increases food insecurity. Water use efficiency is the key factor affecting food insecurity in the short run. The results reveal that the water use efficiency shocks will decrease food insecurity. The shocks experienced by food insecurity can be predicted as self-shock over a span of ten years. Emphasis is given to the task of water management that may support food security in Saudi Arabia through implementing and enhancing the water use efficiency as an integral part of achieving the SDGs in Saudi Arabia.

University, Al-Ahsa, Saudi Arabia, under the grant contract, KFU Research Winter, Grant No.5411. The funder had no role in study design, data collection and analysis, decision to publish, or preparation of the manuscript.

**Competing interests:** We have a potential conflict of interest that the co-author of this article Dr. Mohammed Al-Mahish is one of the Academic Editors of this journal PLOS ONE. This does not alter our adherence to PLOS ONE policies on sharing data and materials.

**Abbreviations:** SDG, Sustainable Development Goal; FIMI, Food Insecurity Multidimensional Index.; BVAR, Bayesian Vector Autoregressive.; AIC, Akaike Information Criterion.; SBC, Schwartz-Bayesian.; HQ, Hannan-Quinn.; IRFs, Impulse response functions.; FVED, Forecasting variance error decomposition.; FAO, Food and Agriculture Organization.; EXR, Exchange rates.; FIES, Food Insecurity Experience Scale.; GLMM, Generalized linear mixed-effect modeling.; WUEA, Water use efficiency for agriculture.; WUEI, Water use efficiency for industries.; WUES, Water use efficiency for services.; AV, Actual value of food insecurity.; FI, Food insecurity.; MIV, Minimum value of food insecurity.; MAV, Maximum value of food insecurity.; UBD, Unknown break date for the variables under consideration.; SBD, Structural break date.; LM, Lagrange Multiplier test for autocorrelation.; LR, Likelihood Ratio.; FPE, Final prediction error.; HQIC, Hannan-Quinn information criterion.; SBIC, Schwarz information criterion.; RMSFE, Root mean squared forecast errors.; ML, Marginal likelihood.; MCMC, Markov chain Monte Carlo..

## Introduction

The recent food crisis increase has attracted the interest and attention of government policy-makers. The FAO defines food insecurity as a situation that occurs when people are in deficit when it comes to securing access to an appropriate amount of safe and nutritious food for normal growth and development, as part of an active and healthy life. Approximately 2.37 billion people did not have access to sufficient food in 2020 and about 148 million people were strictly exposed to food insecurity in 2020, which is more than in 2019 [1]. The recent report issued by [2] shows that in 2021, food insecurity increased and that about 29.3% of the global population was moderately or severely food insecure, with 11.7% facing severe food insecurity. Various factors affect food insecurity at the micro (household) and macro levels (country). At the household level, food insecurity is affected by socioeconomic factors and food, as well as food habits and expenditure [3], while at the country level, food insecurity is affected by the combined effect of population growth, water scarcity, land use change, economic situation, and climate variability [4–7].

The increasing global population and the incidences of poverty, climate change, and low agricultural productivity, among others, are threatening the provision of food, adding to the rapidly rising and volatile food prices of recent decades as significant manifestations of the changes in global food security. On the other hand, the crude oil prices and US dollar exchange rate affects global agricultural products [8] inversely affecting food security. Moreover, [9] examined how food insecurity is affected by the exchange rates and real GDP in exported oil countries. Considering the exchange rate factor (US Dollar—Saudi Riyal exchange rate, one US Dollar = 3.75 Saudi Riyal, on 9 September 2023) that influences food security in a country like Saudi Arabia, it was noted that there were minor changes in the exchange rate during the 1990s - 2000s. As shown in Fig 1, there is a negligible decrease in the exchange rates (EXR) during 2005–2007, becoming constant after and before this era. Also, we can observe that the GDP has been growing faster across 2017–2020. The GDP is likely to be greater in 2020 than in 2009 (economic crisis period) which indicates the further development of the Saudi economy throughout this era.

Despite a dramatic increase in food production, industry, and technologies, food insecurity persists at excessively high levels. Mainly in developing countries, conflicts and economic downturns are the major derivatives of food insecurity [10, 11], whereas in developed countries, a lack of agricultural raw material is the key problem [12]. As expected, desert regions include places where the unfertile soil is lightly unsuitable for agricultural activities and there is an undesirable climate leading to a shortage of crop and livestock products. In the context of Saudi Arabia, there has been a noticeable rise in water demand over time, specifically within the agricultural and industrial sectors [13]. Adding the existing conditions of domestic agricultural production and the resulting products will not be able to fruitfully address the increasing food demand due to the limited fertile land and water resources. Consequently, Saudi Arabia mainly depends on imported food. Moreover, in recent years, the escalating threat of a hotter climate, the deterioration in precipitation, and harsh droughts in Saudi Arabia have reduced the predictability of water management efficiency and resulted in the exhaustion of water bodies and the serious degradation of ecosystems that have gone on to affect the agriculture systems. Groundwater is considered one of Saudi Arabia's most significant and only freshwater resources and is scarce and impaired by aridity, overexploitation, and low rainfall [14]. However, contaminated groundwater can affect the health risk [15] by reducing crop yield directly and threatening food security indirectly. Currently, there is a scarcity of studies in Saudi Arabia that have modeled the factors that affect the FIMI using a robust model such as the Bayesian approach. Our study was established to identify and estimate the FIMI term and aims in

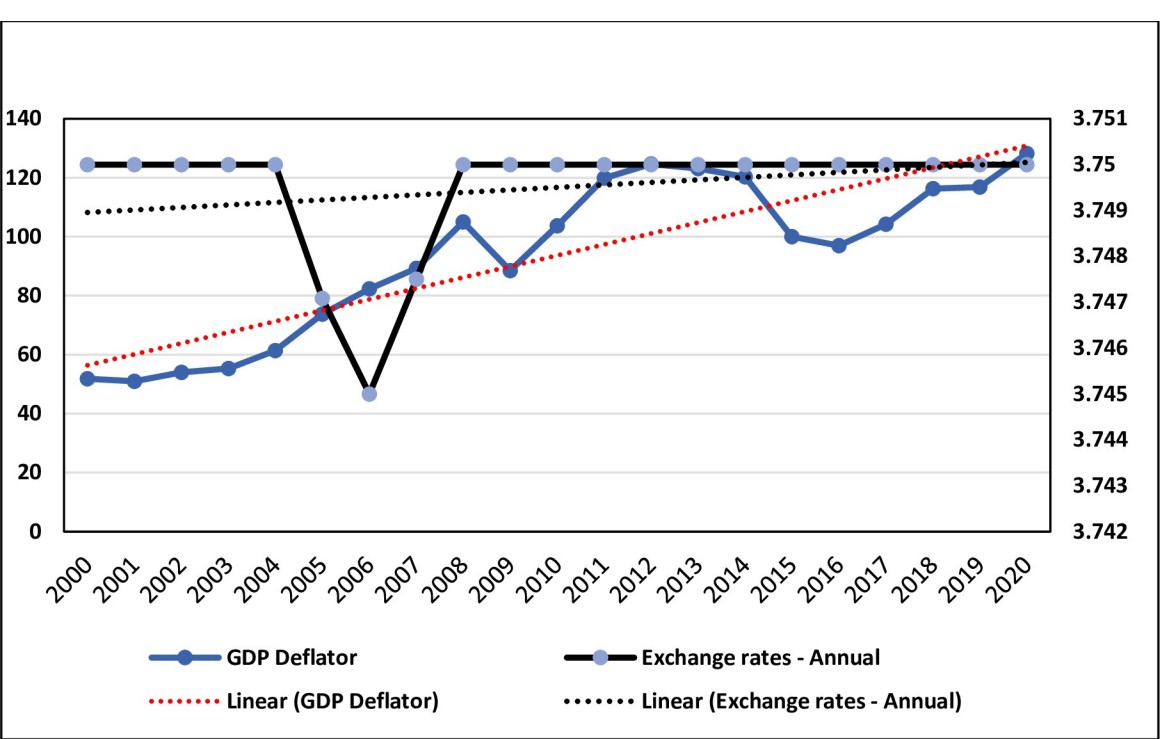

**Fig 1. GDP deflator and exchange rates—annual in Saudi Arabia.** Source: [2] and author's design (2023). Note: (1) Right axis represents EXR values, EXR = 5E-05T + 3.7489, $R^2$ = 0.10, the time trend equation for EXR, indicates that when the time increased by one year the EXR will increase by 0.00005 USD. (2) Left axis represents GDP values, GDP = 3.7235T + 52.695, $R^2$ = 0.7574, the time trend equation for GDP, indicates that when the time increased by one year the GDP will increase by 3.7235 million USD.

order to identify the dynamic effect of sustainable water on the prediction of food insecurity to evaluate the importance of shocks to the water efficiency variables affecting food insecurity in Saudi Arabia. This study also aims to assess the impact of water efficiency on food insecurity in Saudi Arabia. We focus basically on the Bayesian model to answer the research question, what is the likelihood that water efficiency will affect food insecurity in the future?

The importance of the relationship between water use efficiency and food insecurity aligns with the existing literature. The alignment with the existing literature underscores the importance of improving water use efficiency in agriculture specifically to achieve food security [16–19], to mitigate water scarcity [20], to enhance climate change resilience for crop production [21–23] and to promote sustainable development [24]. Scholars as well as international organizations have explored the role of water use efficiency in addressing food security and sustainable agriculture. Some scholars have measured the relation between food insecurity and water use at the household level and ignored the other uses of water. For instance, [25] used multilevel generalized linear mixed-effect modeling (GLMM), to assess the impact of water use in households and found that household water insecurity was positively connected with household food insecurity. [26] used multiple correspondence analysis (MCA) as the categorical data to measure the food insecurity and water insecurity of rural households and found that the food insecurity and water insecurity dimensions are sufficiently distinct to be characterized via separate indicators. Other researchers have focused on water sacristy and food insecurity in agricultural activities and food production and ignore water use efficiency for other purposes [27–30]. The study investigated the crop management plans used for enhancing water use efficiency and stated that there is an urgent need to devise efficient practical crop and soil

management policies to improve water use efficiency in agroecosystems [31]. A recent study in Saudi Arabia focused only on the analysis of water management used to find viable and workable solutions to achieve the sustainable management of scarce water resources, confirming that the primary challenges facing water resource management are the depletion and degradation of surface and subsurface water sources [32]. This study used the Water-Resource Load Index for measuring connection between water resources, population, and economic development and found that the abundance of water resources determined the amount of water for agricultural development and that the urban population increased the water demand. It was concluded that it is a matter of urgency to take effective measures to reduce the amount of water resources used to meet food-security needs [33]. Furthermore, the United Nations Sustainable Development Goals (SDGs) (goal 2 on zero hunger and goal 6 on clean water and sanitation) indicate the significance of water use efficiency as part of achieving food security and sustainable water management [34]. The alignment with these global goals emphasizes the significance of water use efficiency as a critical component of broader development agendas. Publishing a paper on these topics can help inform policymakers, providing them with valuable insights, data, and recommendations. It can contribute to mapping policies, strategies, and programs aimed at addressing food insecurity and water use at the local level. This study considers water use efficiency for various purposes and determined that this study should be performed because achieving a paper on food insecurity and water efficiency can contribute to raising awareness and promote actions to mitigate the challenges associated with food insecurity and water use efficiency.

In the context of the methodology and results contribution of this study to the existing literature, it is twofold. The first, is based on Bayesian Vector Autoregressive (BVAR) forecasting for the purpose of testing a theoretical linkage between two concepts, with FIMI and water efficiency as sustainable development indicators where this is a new study performed in this field. Second, is the enrichment of the generalizability of the literature review through more appropriate outcomes for FIMI and water efficiency. Even though research has been conducted on food security and/or food insecurity in Arab regions, there is a lack of research that investigates the factors affecting FIMI in Arab regions at a macro level, hence this study seeks to fill in this gap related to FIMI in the Middle East. The significance of this study to the international water community can be assessed based on its potential benefits. Conducting this study in arid regions, such as Saudi Arabia, can serve as a benchmark for similar areas facing water scarcity and food security issues. By monitoring the connection between food insecurity and water management shocks, policymakers can gain valuable insights to effectively explore the involved challenges of global water scarcity and food insecurity.

This article is organized into five sections as follows. The second section outlines the review. The third section presents the data and model applied. The fourth section interprets and explains the results and discussion. The last section concludes with the outcomes and recommendations.

## Review

As food insecurity is still a major global challenge [35, 36], several empirical studies have investigated the relationship between food insecurity and prices, water issues, GDP, population, climate changes, and drought, among other factors. Drought is among the major factors affecting food insecurity in the world. A study conducted in the Middle East examined the causal connection between food security and drought, implanting a Bayesian approach to show the significant impacts of drought, agricultural products, and population growth on food security [37]. [38] used a cross-sectional survey of households and applied the Bayesian binary

logistic regression. They found that the significant predictors of food insecurity were socioeconomics, loans, land, and livestock units. A similar study applied a Bayesian logistic framework for evaluating the role of socioeconomic factors on food security, indicating that rural areas face the worst conditions when it comes to food insecurity [39]. In the context of the connection between population and food insecurity, another study examined the impact of population growth on food insecurity by applying an auto-regressive distributive lag (ARDL) co-integration approach. It was confirmed that population growth has a significant impact on food insecurity in the both short-run and long-run [40].

Numerous studies have investigated the analysis of FIMI using household surveys. [41] adopted the multidimensional measurement of poverty in calculating an index of multidimensional food insecurity for South Africa and found that nearly half of the population is considered to be multidimensional food insecure. [42] investigated FIMI for a rural population using the principal component analysis index to evaluate the impact of climate shocks on food insecurity, and found that rainfall and temperature have a significant impact on a household's food security score. Furthermore, [43] used multilevel linear models for identifying the factors of food insecurity in Latin America and the Caribbean using the FAO Food Insecurity Experience Scale (FIES) and confirmed that education, social capital, and GDP per capita result in the largest boost in the probability of food insecurity. [44] applied a mix of quantile and fixed effect models to analyze the determinants of food insecurity and found a strong connection between per capita real GDP and food insecurity, as well as a weak connection between the approach to an enhanced water source and food insecurity.

The food insecurity-water nexus has become a controversial issue. Others argue that water use management efficiency is the key factor of the major shocks facing food security [45] while [46] investigated the link between water and food insecurity. They indicated that direct and indirect water use/ reuse plays a key role in food security. [25] applied the Tobit Mixed Effects regression model to measure the relationships between income and both water expenditure and insecurity. They found a positive association between water expenditure and food insecurity. Moreover, [47] applied multilevel generalized linear mixed-effect modeling (GLMM) and found that water insecurity is the driver of food insecurity.

Concerning the application of BVAR for analyzing economics research, [48] used BVAR to estimate the influence of economic policy uncertainty on stock market revenue and found that stock market returns have been negatively influenced by the increased economic policy uncertainty levels. In contrast, [49] applied a Bayesian time-varying parameter VAR model with stochastic volatility in order to explore the dynamic connections of GDP growth and energy use. It was observed that the GDP has a positive shock of energy use in diverse time horizons. Likewise, very recently, [50] used a combination of impulse response functions (IRF) and variance error decomposition (FVED) and applied a panel VAR model to examine the response of global food prices to various variables. It was indicated that global food prices decrease with climate-friendly agricultural technologies and increase with an increasing global surface temperature.

From the literature review, we concluded that food security and/or insecurity emphasizes and addresses several econometrics techniques. It has been analyzed by applying dissimilar econometric models to various datasets covering cross-section, panel, and time series data. Despite remarkable investigations into food insecurity and its fundamental challenges, no article has linked water management efficiency and food security using the BVAR system.

## Data and models

To verify the framework's faithfulness and the reliability of the data, we chose the United Nations Agencies as the main sources of data, for instance, FAO and the World Bank. The data used in this study principally depends on the data of the food security indicators for measuring FIMI expressed as a percent. From the literature, various variables might influence one country's food insecurity level, including economic, social, political, and sustainability variables. In our study, the data on the factors able to predict the effect of FIMI was collected covering the information on water use efficiency represented as an SDG factor. We integrated the SDG in this study because [2] pays attention to SDG and food security. Besides this, the most recent developments highlight the significance of sustainability development which may be considered a long-term time dimension (fifth) in relation to food security [51]. Besides, water is not only part of various other SDGs but in several aspects, it is a key precondition of many goals [52]. The water information used in this study includes the water use efficiency for agriculture (WUEA), industry (WUEI), and services (WUES), measured using US$/m3. The reference period for the data extended from 2000–2020. The restriction in terms of period is related to the availability of water use efficiency data from the year 2000. We were interested in data on water use efficiency for several reasons. Essentially, water is considered to be a key factor in food insecurity in Saudi Arabia [53]. Likewise, the water variables were initially driven by the literature as one of the determining factors of food insecurity.

The shocks to the water efficiency variables affecting the food insecurity of this study were achieved across three sequent stages using the Stata and R- studio software: (1) the estimation of the FIMI model, (2) the estimation of the VAR model, and (3) the estimation of the BVAR model. In our study, we applied the reduced form VAR predictable using the Bayesian method since it is perfectly appropriate for shorter datasets. The steps and models designed for the study were illustrated in Fig 2.

## FIMI model

For building FIMI for Saudi Arabia, we used the four dimensions of food security generated by [54] availability, access, utilization, and stability. This stage follows the previous estimation of FIMI conducted by [55] and the recent study by [56] based on the neutral weights approach. First, each variable needs to be normalized according to its max and min values. Normalization is fundamental for any aggregation as the indicators in a dataset have unique units. Aggregation is significant where the indicators are comparable [57]. For this study, min, max transformations, and rescaling are approved. Max-min normalizes the indicators so then they have an identical range (0,100) by subtracting the min value and dividing it by the range of the indicator values. Hence, the normalization for some series variables was calculated based on the min and max data values appearing in the dataset for each assumed year. The normalization equation used the following formula:

$$FIMI = \frac{100*(AV - MIV)}{(MAV - MIV)} \quad (1)$$

Where AV = actual value of food insecurity (FI), MIV = minimum value of FI, MAV = maximum value of FI. For establishing the FIMI term, we used the familiar food security dimension variables which were built based on the coupling of [55]. According to [58], FIMI scored from 0 (best score) to 100 (worst score). The Food Insecurity Map countries were divided into four groups based on the final FIMI score: moderate food insecurity (30<FIMI<39.99, serious food insecurity (40<FIMI<49.99), alarming food insecurity (50<FIMI<59.99), and extremely alarming food insecurity (FIMI>60). Based on [58]

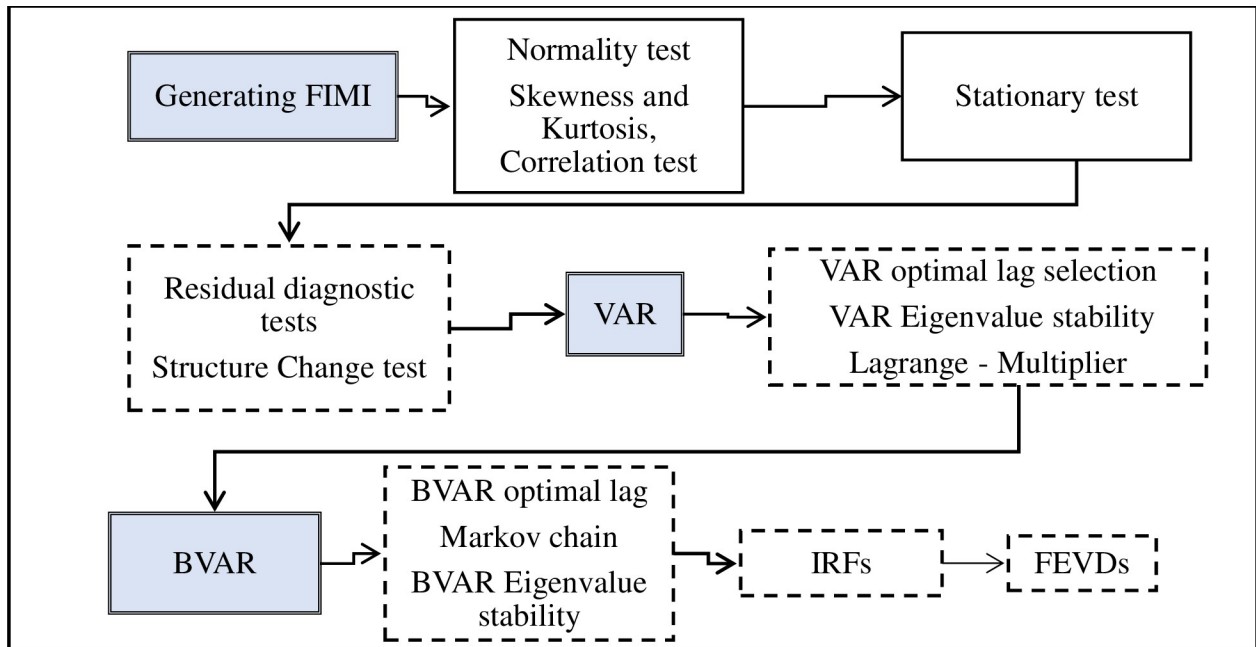

**Fig 2. Flowchart of the analytical methodology. Source**: Authors' design, (2023).

estimations, Saudi Arabia ranked 41 with an FSI score of (69.9) with an improved score of (11.8) in 2022.

### Zivot and Andrews structural-break test

The well-known Dickey-Fuller (DF) [59] and Phillips and Perron (PP) [60] unit root tests are biased [61] and do not allow for the possibility of a structural break. Therefore, before the VAR and BVAR model estimations, structural break unit root tests were used for testing the integration rank of each variable for the Unknown Break Date (UBD) for the variables under consideration. As suggested by [62, 63], a UBD of the critical values was obtained by applying a break date estimated using the minimum t-statistic for the unit. [62] proposed three structural unit root models, A, B, and C. The three models display a unit root methodology that allows for the depicting of a single structural break and establish the treatment of a structural break as an endogenous rather than exogenous trend. Therefore, (1) model A permits a change in intercept, i.e. a change in the level of the series; (2) model B permits a change in trend (for a one-time change in the slope of the trend function), and (3) model C allows for a change in both the intercept and slope (one-time changes in the level and slope of the trend function of the series). To test for a unit root against the alternative of a one-time structural break, in this study, we applied the Zivot and Andrews [62] regression equations corresponding to models A, B, and C and used the following equations:

$$\Delta Y_t = K + \alpha Y_{t-j} + \beta_t + \gamma_i UD_t + \sum\nolimits_{j=1}^{k} d_j \Delta Y_{t-j} + \varepsilon_i \quad \text{Model (A)} \tag{2}$$

$$\Delta Y_t = K + \alpha Y_{t-j} + \beta_t + \theta UD_t + \sum_{j=1}^{k} d_j \Delta Y_{t-j} + \varepsilon_i \qquad \text{Model (B)} \qquad (3)$$

$$\Delta Y_t = K + \alpha Y_{t-j} + \beta_t + \theta UD_t + \gamma_i DT_t + \sum_{j=1}^{k} d_j \Delta Y_{t-j} + \varepsilon_i \qquad \text{Model (C)} \qquad (4)$$

Whereas $\Delta$ is the first difference, $Y_t$ denotes the variable series containing a unit root refer-ring to the existing studies of FIMI, WUEA, WUEI, and WUIS. The $Y_{t-j}$ terms on the right-hand side of the three equations permit serial correlation and confirm that the disturbance term is white noise with variance $\sigma^2$, and t = 1...., T which denotes to index of time. $UD_t$ is an indicator dummy variable for a mean shift occurring at each possible time break date (TBD) while $DT_t$ is the corresponding trend variable, whereas:

$$UD_t = \begin{cases} 1 \; if \; t > TBD \\ 0 \; otherwise \end{cases} \qquad (5)$$

and

$$TD_t = \begin{cases} 1 - TBD \;\; if \; t > TBD \\ 0 \qquad\qquad\; otherwise \end{cases} \qquad (6)$$

The null hypothesis ($H_0$) of the three models is $\alpha = 0$, which indicates the presence of unit root in series ($Y_t$) with drift that excludes any structural break, whereas the alternative hypoth-esis $\alpha < 0$ indicates that the series is trend stationary. Most scholars have applied Model A and/ or C and, according to previous studies [64–66], we applied Model C for the analysis of the unit root as it allows for a break in both intercept and trend. Model C is also more compre-hensive than the A and B models.

## Conventional model

We used the conventional equation model in which a single dependent endogenous variable was determined by one or more exogenous explanatory variables. To achieve the aims of this study, a predictive model was derived as follows:

$$Y_t = \beta_o + \beta_{1-k} w_{1-k,t-i} + \varepsilon_i \qquad (7)$$

Where Y denotes the endogenous variable (FIMI) at time t in this study, and $w_{,t-i}$ denotes the water efficiency use at lag (1 to k) of predictive exogenous variables.

## Vector autoregressive (VAR) model

The VAR model was primarily proposed by [67] and has been broadly applied in the analysis of macroeconomics. In this study, before the estimation of the BVAR model proposed by [68], we considered a VAR model that uses a reduced-simultaneous form as follows:

$$Y_t = C_o + \sum_{I=1}^{K} \beta_i Y_{t-i} + \varepsilon_t \qquad (8)$$

Where $Y_t$ is the vector of the endogenous variables (n x1) in our study n, specifically FIMI, WUEA, WUEI, and WUIS to be forecasted. The only deterministic component is a constant term denoted by $C_o$ as a constant term (n x1), a vector of the intercept. $\beta_i$ is the matrix of coeffi-cients for the $i^{th}$ lag (n x n) polynomial matrix in the backshift operator with lag length p, and

$\varepsilon_t$ (n x 1) vector of white-noise error terms, i.e. the vector comprising the reduced form residuals, which in general will have non-zero correlations. Eq (8) is estimated by ordinary least square (OLS) including three lags. We applied the lag length selection criteria for selecting the number and chosen lags according to the explicit statistical information criterion.

## The lag length selection criteria

After the VAR analysis, the selection of the length lag is very important for determining the lag length for the VAR(s) model using the optimum model selection criteria. Confirming [69, 70] for establishing the equations of the selection lag criteria used to determine the appropriate lag length, the Akaike Information Criterion (AIC), Schwartz-Bayesian (SBIC) criterion, and Hannan-Quinn (HQIC) criteria were used [71]. In our study, we applied three selection information criteria, AIC, SBIC, and HQIC. We calculated the following formulas for each lag length criterion:

$$AIC = \log[\Sigma] + \frac{2g}{n} \tag{9}$$

$$SBIC = \log[\Sigma] + \text{glog}\frac{glog(n)}{n} \tag{10}$$

$$HQIC = \log[\Sigma] + \text{glog}\frac{2loglog(n)}{n}g^2 \tag{11}$$

Where $|\Sigma|$ is the determinant of the variance-covariance matrix of the residual of the system, g is the total number of parameters estimated in all equations, and n is the number of observations.

Thus, if we have m equations and p lags, the intercept in each equation can be written as:

$$g = m^2\text{p} + \text{m} \tag{12}$$

We approved the model selection according to the lowest AIC or SBIC value. From the literature, the VAR model utilizes equal lag length for all variables of the model. One deficiency of the VAR model is that several parameters are required to be estimated, some of which may be insignificant. This leads to an over-parameterization problem resulting in multicollinearity and deficit degrees of freedom when making the inefficient estimates [72]. Therefore, [73] suggested that an alternative approach to avoid this over-parameterization is to use a BVAR model.

## Bayesian vector autoregressive (BVAR) model

As the study of food security has been estimated by FAO since (1990) and there is no data available for most food insecurity indicators for Saudi Arabia after 1990, the weakness of maximum likelihood estimation in the small sample can be solved by Bayesian estimation as an alternative technique. Uncertainty is important to estimate for economic variables. This estimation solves the challenge of the assumption of the VAR approach since it is flexible [74]. Furthermore, BVAR models generally produce more accurate forecasts. We approved the BVAR approach, Eq 13, following the examples of [75, 76] which takes the following form:

$$Y_t = C_o + \beta_i Y_{t-i} + - - - \mp \beta_p Y_{t-\text{p}} + \varepsilon_t \; with \; \varepsilon_t, N(0, \; with \; \Sigma). \tag{13}$$

Where $Y_t$, denotes our interested 4 endogenous variables, FIMI, WUEA, WUEI, and WUIS, a 4 x1 column vector in the BVAR system, while $C_o$ denotes a 4 x 1 vector of the

intercept. $\beta$i, (i = 1-----p), denoting a 4 x 4 matrix of autoregressive coefficients of regressors; p is the order of the BVAR and, finally, $\varepsilon$ is a 4 x 1 vector of Gaussian exogenous shocks with a zero mean and variance-covariance (VCOV) matrix $\Sigma$. The $4 + 4^2 p$ is the number of coefficients to be estimated that increase quadratically with the number of involved variables and linearly in the lag order [75].

We used Markov chain Monte Carlo (MCMC) values, which are commonly applied for fitting Bayesian statistical models. MCMC was then summarized as means or medians where a single statistic is required [77]. By default, the command calculates the posterior means, posterior standard deviations, and 95% equal-tailed credible intervals for the forecast for every result variable. The study also checked that the MCMC converged before proceeding with further analyses by performing graphical checks.

Finally, Bayesian forecasting analysis was performed using the impulse–response functions (IRFs) and forecast-error variance decompositions (FEVDs), which are commonly used to describe the results from multivariate time-series models such as the VAR model [78]. Hence, one of the ways to investigate the dynamics of a model is to focus on the impulse response functions (IRFs) or FEVDs for estimating the progress of the variable shocks. VAR analysis often leads to the estimation of IRFs and FEVDs which are essential parts of the VAR method. We followed the example of [79, 80] for setting the IRFs and FEVDs for a 10-year horizon.

In our study, IRFs measure the effect of a shock to one variable such as water use efficiency on a given interesting variable such as food insecurity. In this study, the Bayesian IRFs were computed from the mean posterior distribution of IRFs to provide more stable estimates for the datasets.

The impulse response of the orthogonalized IRFs in the VAR model examine the sensitivity of the dependent variable to shocks to each of the variables [81]. Therefore, the impulse response at horizon h of the variables to an exogenous shock to variable y can be easily displayed using the Choleski decomposition (Eq 14) proposed by [67] as follows:

$$y_t = \sum_{i=0}^{\infty} \emptyset_i v_{t-i} [\emptyset_0 = I_k \text{ is the } (KxK) \text{ identity matrix}] \tag{14}$$

Whereas $\emptyset_i$

$$\emptyset_i = \sum_{j=1}^{i} \emptyset_{i-j} A_j [i = 1, 2, 3, \ldots.] \tag{15}$$

Where $\emptyset$i are explained as impulse responses of the model; Aj = 0 for j >p (for a k dimensional VAR (p) process), and $v_t$ represents the orthogonal residuals [82]. In addition, the IRFs have no causal interpretation but they measure the probability of a shock to one variable impacting the other variables. In addition, the decomposition is not unique but influenced by the ordering of the variables [83]. Variance decomposition explains the fraction of changes in the dependent variable due to their shocks [84].

The h-step ahead forecast error equation used in this study is written as:

$$Y_{it+h} = E[Y_{it+h}] = \sum_{k=0}^{h-1} A_j \Big[ e_{i(t+h-i)} \emptyset_i \tag{16}$$

Where $Y_{it+h}$ is observed, vector at time t+h, $E[Y_{it}+h]$ is the h-step ahead predictor vector made at time t or the orthogonalized shocks $E[Y_{it}+h]$ are the h-step ahead predictor and the g-step ahead predictor vector made at time t; the orthogonalized shocks $e_{it}K^{-1}$ (with K matrix) have a covariance matrix $l_k$ [83]. In our study, we used the medians of the FEVDs.

**Empirical results and discussions.** This section presents a summary of the descriptive results, as well as the heteroskedasticity test, unit root tests; and the estimation results of the VAR and BVAR models.

**Descriptive results.** Table 1 shows that the average IFMI for the 21 years was 21.02%, with a maximum of 22.86% and a minimum of 18.94%. WUEA, WUEI, and WUIS averaged US$/m3 0.78, US$/m3 373.36; and US$/m3 88.91 respectively. The skewness statistics for FIMI show as being negatively skewed (-0.16) since it is less than zero, denoting that most of its values fall on the left-hand side of the mean, whilst WUEA, WUEI, and WUIS are positively skewed since their skewness statistics are greater than zero at 0.36, 0.85 and 0.45, respectively.

Furthermore, the kurtosis value, whose threshold is less than three, indicates that the FIMI (1.90), WUEA (2.11), and WUIS (2.03) variables are platykurtic distributed, except for WUEI, which is leptokurtic distributed with a threshold of more than three (4.79). The correlation analysis result obtained from Table 1 indicates a negative correlation between WUEI and WUIS (r = -0.80, p < .001).

## Estimation of FIMI

The radar chart (Fig 3) shows that the FIMI for Saudi Arabia is estimated to be less than 23.00 over the study period (between 22.86 in 2011 and 18.94 in 2007) which implies that no food insecurity was observed in Saudi Arabia throughout this span of the study (less than moderate food insecurity group, 30<FIMI<39.99). Our results agreed with those of Hassen and [85] who indicated that the six GCC members are food-secure in the Arab world and among the most food-secure countries in the world.

## Zivot–Andrews, structural unit root and residual diagnostic tests

The results of the Zivot–Andrews unit root test with BD for each time series are reported in Table 2. This unit root test suggests one BD in the variables. The null hypothesis for Zivot-Andrews is the presence of a unit root with a structural change in intercept, trend, or both. As shown in Table 2, the null hypothesis cannot be rejected when the critical values of (1%, 5%, and 10%) are greater than the test statistic value.

The unit root test endogenously recognizes the point of the specific most significant structural break date (SBD) in each time series observed. In the case of both intercept and trend, the results show that the FIMI and WUEI series studied manifested the existence of a structural break in 2013. In contrast, the test identified a break in the WUEA and series WUIS in 2015 and 2017, respectively. The results also revealed that all tested variables were non-stationary at the level but stationary at the first difference. Thus, it is more likely that Eq (8) can be used as an acceptable model for forecasting.

Furthermore, the structural breaks were tested. We implemented F-statistics to test for potential change points in the dataset. The null hypothesis of no structural change was rejected at the one percent level. Also, the table shows potential change points for FIMI, WUEA, WUEI, and WUIS, corresponding to years 2010; 2008; 2002 and 2014, respectively.

The study examines both the robustness of the heteroskedasticity [86] and the [87] tests for investigating the residual diagnosis (Table 2). The results demonstrate a serial correlation between the variables and the studied variables show a normal distribution except the WUEI (P- value = 0.07), which were transformed into natural logarithms (Ln) accordingly.

## Estimation of the VAR model results

Before exploring BVAR analysis, the conducting of proper VAR analysis was important to identify the appropriate lag period that should be applied. We applied econometric models to

**Table 1. Descriptive statistics of the selected variables.**

| Variable | FIMI | WUEA | WUEI | WUIS |
|---|---|---|---|---|
| Mean | 21.02 | 0.78 | 373.36 | 88.91 |
| Std. dev. | 1.23 | 0.13 | 98.08 | 15.07 |
| Min. | 18.94 | 0.85 | 222.21 | 69.35 |
| Max. | 22.86 | 0.99 | 661.24 | 118.21 |
| Skewness | -0.16 | 0.36 | 0.85 | 0.45 |
| Kurtosis | 1.90 | 2.11 | 4.79 | 2.03 |
| Matrix correlation | | | | |
| FIMI | 1.00 | | | |
| WUEA | 0.20 | 1.00 | | |
| WUEI | 0.03 | 0.17 | 1.00 | |
| WUIS | 0.26 | 0.14 | -0.80*** | 1.00 |
| Observation | 21 | 21 | 21 | 21 |

*** $p < .001$.

Source: FAO (2022) and authors' calculations (2023).

ascertain the most suitable lag period that best fitted the analysis, given the interesting data. The result indicates lag order 2 as selected by the VAR lag order selection criteria. Therefore, noticeably, lag 2 is the most appropriate for the estimation, hence lag was chosen based on the level that has the most *(asterisk) among all the criteria (implies the highest criteria preference).

Moreover, the stability of the VAR model was checked using the eigenvalue and the modulus results indicate that the model is stable (less than one), therefore the results of the analysis can be ascertained as being reliable, confirming the stationarity condition of the variables.

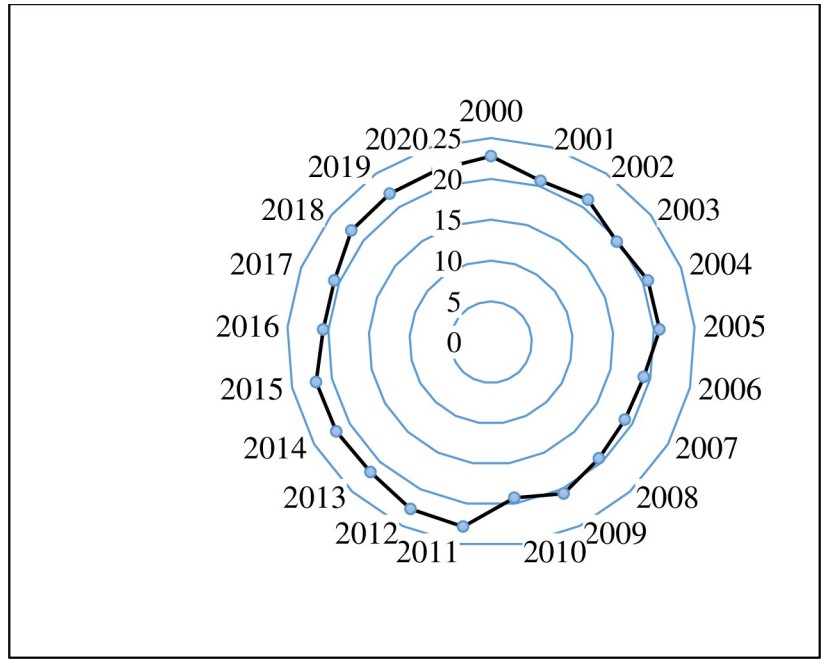

**Fig 3. FIMI in Saudi Arabia (2000–2020). Source**: Authors' design, (2023).

**Table 2. Zivot–Andrews's unit root and residual diagnostic results.**

| Variables | Intercept* | | Trend** | | Both*** | |
|---|---|---|---|---|---|---|
| | t-Stat | BD | t-Stat | BD | t-Stat | BD |
| FIMI | -7.558 | 2013 | -6.071 | 2012 | -7.097 | 2013 |
| WUEA | -4.807 | 2017 | -4.542 | 2017 | -4.614 | 2015 |
| WUEI | -4.882 | 2013 | -4.886 | 2004 | -4.371 | 2013 |
| WUIS | -4.845 | 2017 | -4.281 | 2016 | -4.838 | 2017 |

\* The critical values for the Zivot and Andrews test are -5.34, -4.80, and -4.58 at 1%, 5%, and 10% levels of significance: respectively.
\*\* The critical values for the Zivot and Andrews test are -4.93, -4.42, and -4.11 at 1%, 5%, and 10% levels of significance: respectively.
\*\*\* The critical values for the Zivot and Andrews test are -5.57, -5.08, and -4.82 at 1%, 5%, and 10% levels of significance: respectively.

| **Structure Change test** | | | |
|---|---|---|---|
| Variable | F-Statistics | $H_0$:No Structural Change | Potential Change Points |
| FIMI | 25.919 | Reject $H_0$ | Year 2010 |
| WUEA | 64.839 | Reject $H_0$ | Year 2008 |
| WUEI | 16.581 | Reject $H_0$ | Year 2002 |
| WUIS | 29.682 | Reject $H_0$ | Year 2014 |

| **Residual diagnostic tests** | | | |
|---|---|---|---|
| | **FIMI** | **WUEA** | **WUEI** | **WUIS** |
| Heteroskedasticity: Breusch–Pagan's test | 0.58 (0.31) | **0.18 (0.67)** | 0.25 (062) | 0.33 (0.56) |
| Normality: Jarque-Bera test | 1.16 (0.56) | 1.15 (0.56) | 5.26* (0.07) | 2.04 (0.36) |

chi2 = 4.288, Prob > chi2 = (0.04) **, $H_0$: no serial correlation. Durbin–Watson d-statistic (7, 21) = 2.847827.
Serial correlation: Breusch–Godfrey LM test for autocorrelation

\*\*, \* Levels of significance at 5% and 10%; respectively.
Source: Authors' calculations (2023).

The predicted error is estimated as a mean (-3.14e-09), which is very close to zero. This shows that the variables are normally distributed. In addition, a further post-estimation test for the LM test was performed, and it is obvious that from findings the p-value is greater than 5% for the LM test (probability values of the two lag orders estimated as 0.17 and 0.26), indicating that we fail to reject $H_0$, hence there is no autocorrelation in lag order.

In the econometric investigation, the VAR models investigated where each dependent variable was regressed on its self-lags and the lags of the other variables. Here, the variable performed as a dependent and independent variable. The results of the VAR analysis obtained from the outcomes display that in the short run, the water use efficiency for agriculture and services has a significant impact on food insecurity in Saudi Arabia. This means that in the short-run term, changes in water use efficiency have a explanatory power over the current FIMI. This make senses because it is a key factor affecting food insecurity in the short run. Improving water use for agricultural purposes will reduce food insecurity [16]. On the other hand, the response of food insecurity to water use efficiency for industrial purposes is largely insignificant.

## Estimation of the BVAR model results

The results of the BVAR analysis indicate that in the short run, all coefficients of water use efficiency are significant on the FIMI. This proves that food insecurity is better explained by water efficiency for various purposes. However, the root means square forecasting error for WUEA is less than one in comparison with WUEI and WUIS. This implies that the BVAR model provides a better forecast with interdependence for WUEA and FIMI. Hence, the BVAR model with the minimum RMSE emerged as the preferred model (Table 3).

**Table 3. BVAR model results.**

|  | FIMI | WUEA | Ln.WUEI | WUIS |
|---|---|---|---|---|
| FIMI (-1) | .3614831 [.092284] (3.92) *** | 0132755 [.0115657] (1.15) | 9.522286 [8.152015] (1.17) | 2.938158 [1.922798] (1.53) |
| FIMI (-2) | 4631685 [.091406] (5.07) *** | -.0117534 [.0114557] (-1.03) | -.9773224 [8.074456] (-0.12) | .2153612 [1.904505] (0.11) |
| FIMI (-3) | .2502664 [.1028979] (2.43) *** | -.0139561 [.0128959] -(1.08) | 8.532638 [9.089605] (0.94) | .1928715 [2.143945] (0.09) |
| WUEA (-1) | 1.702765 [4.329408] (0.39) | -.2450552 [.5425926] (-0.45) | 844.1862 [382.4432] (2.21) ** | -133.2213 [90.20605] (-1.48) |
| WUEA (-2) | 27.60994 [5.503283] (5.02) *** | .4424262 [.6897112] (0.64) | -755.3004 [486.1388] (-1.55) | 45.01443 [114.6645] (0.39) |
| WUEA (-3) | -25.59556 [3.00458] (-8.52) *** | -.1567479 [.3765557] (-0.42) | 236.7817 [265.413] (0.89) | -3.124537 [2.6024] (-0.05) |
| Ln.WUEI (-1) | -.0056417 [.0033299] (-1.69) * | .0004138 [.0004173] (0.99) | .2274358 [.2941495] (0.77) | .0231056 [.0693804] (0.33) |
| Ln.WUEI (-2) | -.0132361 [.0032985] (-4.01) *** | .0000215 [.0004134] 0.05 | -.1421327 [.29138] (-0.49) | -.1004897 [.0687272] (-1.46) |
| Ln.WUEI (-3) | -.0130246 [.00336] (-3.88) *** | .0012308 [.000421] (2.92) *** | -.5837543 [.296812] (-1.97) ** | .1502078 [.0700085] (2.15) ** |
| WUIS (-1) | -.0775661 [.0303234] (-2.56) *** | .0083653 [.0038003] (2.20) *** | -7.116399 [2.678649] (-2.66) *** | 1.48079 [.6318072] (2.34) *** |
| WUIS (-2) | -.1476708 [.0314636] (-4.69) *** | -.002702 [.0039432] (-0.69) | 4.147908 [2.779371] (1.49) | -.5168511 [.6555642] (-0.79) |
| WUIS (-3) | .0486816 [.0198144] (2.46) *** | .0044908 [.0024833] (1.81) | -5.680971 [1.75033] (-3.25) *** | .4282121 [.4128466] (1.04) |
| RMSFE | .56152 | .070374 | 49.6025 | 11.6996 |
| R-squared | 0.9411 | 0.8266 | 0.8586 | 0.8131 |
| Chi2 | 287.5289*** | 85.8121*** | 109.3344*** | 78.28494*** |

The test statistic (z) is in parentheses, [stad .err.] in square brackets. RMSFE: Root Mean Squared Forecast Errors.

Source: Authors' calculations (2023).

Table 4 presents the lag length results. The outcome shows that the model with three lags (Lag3) has the highest posterior probability (ML) of the three considered models.

## Markov chain Monte Carlo (MCMC) diagnosis

Unlike classical forecasting and frequentist analysis, a Bayesian forecast at a certain time corresponds to not just one value but a sample of (MCMC) values, here combined posterior distributions in a Bayesian analysis, as mentioned. MCMC convergence was checked visually before proceeding with the analysis. The graphics in Fig 4A–4C provide the trace plots due to fitting the BVAR model, revealing that the trace plot does not exhibit any trends and that the autocorrelation is very low. In return, the MCMC appears to have converged and shows that the mean of the Markov chain has stabilized and appears constant over the graphs.

As shown in Table 5, the 95% credible intervals for the individual eigenvalue modulus do not include estimates greater or equal to one, which is a good indication. The posterior probability that all eigenvalues in the unit circle are close to one. We have no reason to believe there is a violation of the stability assumption (Table 5).

**Table 4. Optimal lag selection criteria of the BVAR model.**

| Lags No. | log(ML) | P(M) | P(M|y) |
|---|---|---|---|
| lag1 | -201.5947 | 0.3333 | 0.0000 |
| lag2 | -193.2726 | 0.3333 | 0.0002 |
| lag3 | -184.8788 | 0.3333 | 0.9998 |

Note: Marginal likelihood (ML) computed using Laplace–Metropolis approximation.

Source: Authors' calculations (2023).

## Impulse–response functions

Since BVAR models involve a lot of regression coefficients, it becomes complicated to interpret the outcomes. Instead of individual coefficients, IRFs are applied to summarize. IRFs estimate the influence of a shock in one variable, which is an impulse variable, on a given response-interested variable at a specific time. In our study, IRFs have been used to evaluate the influence of water management efficiency on the other outcomes of IMFI in the model. Here, we used the model with the three lags selected in the previous section.

Table 6 exhibits the responses of food insecurity to shocks to water use purposes. The results obtained from IRFs show there to be a significant effect due to water efficiency on FIMI, i.e., WUEA and WUEI reduce FIMI over the 10-year horizon. Our results agreed with [88] and confirm the negative effect of irrigation interventions on food insecurity. WUIS has increased FIMI after the first period. This indicates that when water use for service purposes increases, food insecurity will also increase (Table 6).

## Forecast error variance decompositions

The FEVD is reported for a 10-year horizon as shown in Table 7. Regarding the outcomes from the variance decomposition of food insecurity, the results verify that a shock to FIMI accounts for the (100%) variation in the fluctuation of FIMI, which means there is a self-explanation. Moreover, the results reveal that the shocks of FIMI were mostly as its self-shock decreases across the selected periods, while water use efficiency for agriculture was the main driver of food insecurity followed by water use efficiency for industrial and services purposes respectively. [16, 89] confirmed that water markets increase water use efficiency in agriculture and it is supposed to play a vital role in ensuring food and water security. The results from Table 7 imply that in period 10, food insecurity is estimated to be about 77% of its self-shocks. In contrast, water use efficiency for agricultural purposes appears to be the highest (67%)

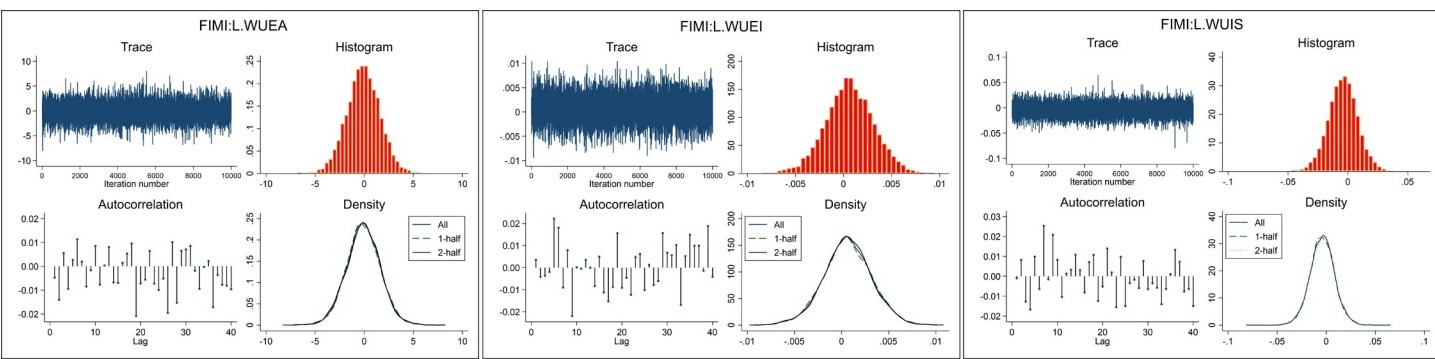

**Fig 4.** A. FIMI:L.WUEA. B. FIMI:L.WUEI. C. FIMI:L.WUIS. **Source**: Authors' design, (2023).

**Table 5. Eigenvalue stability condition of the BVAR model.**

| Eigenvalue | Mean | Std. dev. | MCSE | Median | Equal-tailed 95% cred. Interval | |
|---|---|---|---|---|---|---|
| modulus | | | | | | |
| 1 | .548102 | .0732379 | .000732 | .41932 | .925253 | .812585 |
| 2 | .964383 | .0651191 | .000651 | .9634072 | .8376798 | .1092789 |
| 3 | .8783838 | .0787049 | .000787 | .8867225 | .7038146 | .1012171 |
| 4 | .7346137 | .1276713 | .001277 | .7539166 | .4763001 | .9348375 |
| 5 | .3700557 | .1185975 | .001186 | .3339341 | .2207115 | .6437148 |
| 6 | .285765 | .0555104 | .000555 | .2780882 | .1970691 | .4152818 |
| 7 | .2405682 | .0406321 | .000406 | .2374934 | 1691157 | .3285339 |
| 8 | .2194007 | .0366586 | .000367 | .2187404 | .1485541 | .2939359 |
| 9 | .188631 | .0365472 | .000365 | .1889509 | .11682 | .2606606 |
| 10 | .1706169 | .0372104 | .000372 | .1719407 | .0937563 | .2409223 |
| 11 | .1302215 | .043438 | .000434 | .1317215 | .0432278 | .2102185 |
| 12 | .105433 | .0485498 | .000485 | .1082508 | .0114209 | .1941587 |

Source: Authors' calculations (2023).

Pr (eigenvalues lie inside the unit circle) = 0.9685.

among the other two uses, explaining the variance in food insecurity, while water use efficiency for industrial and services purposes only explained about 49% and 36% of the variance in food insecurity. The results also reveal that across the 10 years, the water use efficiency shocks will decrease food insecurity but at an insignificant rate. This implies that water use efficiency has a low percentage of effects on food insecurity in Saudi Arabia. Hence, we can suppose that water management did not contribute much during the 10 periods of the FEVD; therefore, the shocks in food insecurity could be considered self-shocks during the 10-year period. Naturally, the water resources in the country are scarce.

When comparing the results to those of other studies, it was observed that several studies have confirmed that water use efficiency for agriculture and industrial purposes will increase food security [16, 18, 90, 91]. Also, [92] confirmed that when water is reallocated from agriculture for other uses, it can reduce crop production and increase food insecurity.

## Conclusions

Due to the limited availability of fertile land and water resources in Saudi Arabia, the current state of domestic agricultural production and products is unable to meet the rising population demand. As a result, the country heavily relies on imported food to satisfy its food requirements. This study was aims to identify and estimate the FIMI term by identifying the dynamic effect of sustainable water management on food insecurity and to evaluate the consequence of shocks to water efficiency management in relation to food insecurity in Saudi Arabia. The reference period of the data extended from 2000–2020. The food insecurity prediction was achieved across three sequent stages: (1) the estimation of the FIMI model, (2) the estimation of the VAR model, and (3) the estimation of the BVAR model. We used both impulse response functions and variance error decomposition through 1000 Monte Carlo simulations according to the Bayesian Vector Auto-Regressive (BVAR) model to examine both the response of food insecurity to shocks in water efficiency use for various purposes and the decomposition of error variance in food insecurity.

Food insecurity was not detected in country at any time during the studied period. The variable series manifest the existence of a structural break in different years, and a serial

**Table 6. Bayesian impulse response functions for FIMI to water efficiency.**

| periods | FIMI—WUEA | | | FIMI -WUEI | | |
|---|---|---|---|---|---|---|
| | irf | Lower | Upper | irf | Lower | Upper |
| 1 | -.416778 | -3.88745 | 3.15963 | -.000112 | -.005223 | .005015 |
| 2 | -.699837 | -7.1746 | 5.63784 | -.000999 | -.010863 | .008842 |
| 3 | -1.19691 | -10.5687 | 7.66179 | -.0019 | -.016286 | .012429 |
| 4 | -1.63268 | -13.5836 | 9.4758 | -.002696 | -.021882 | .016264 |
| 5 | -2.00432 | -16.6295 | 11.1256 | -.003416 | -.02781 | .020133 |
| 6 | -2.32043 | -19.5747 | 12.9968 | -.004081 | -.033997 | .024264 |
| 7 | -2.59468 | -22.6667 | 14.4599 | -.004713 | -.040725 | .029362 |
| 8 | -2.84049 | -25.9507 | 16.2168 | -.005333 | -.04867 | .034465 |
| 9 | -3.07193 | -30.1577 | 18.5008 | -.00596 | -.057024 | .039567 |
| 10 | -3.3029 | -34.4578 | 20.9465 | -.006613 | -.066409 | .045477 |
| periods | FIMI—WUIS | | | Posterior means reported. | | |
| | irf | Lower | Upper | 95% equal-tailed credible lower and upper bounds were reported. | | |
| 1 | -.001005 | -.02515 | .023058 | (1) irfname = birf, impulse = WUEA, and response = FIMI1. | | |
| 2 | .000282 | -.046742 | .046173 | (2) irfname = birf, impulse = WUEI, and response = FIMI1. | | |
| 3 | .001296 | -.066068 | .066809 | (3) irfname = birf, impulse = WUIS, and response = FIMI1 | | |
| 4 | .002101 | -.086367 | .089216 | | | |
| 5 | .002816 | -.107317 | .109181 | | | |
| 6 | .003498 | -.128505 | .130323 | | | |
| 7 | .004187 | -.151187 | .15446 | | | |
| 8 | .004915 | -.177327 | .180974 | | | |
| 9 | .005695 | -.203287 | .209067 | | | |
| 10 | .006532 | -.234438 | .243431 | | | |

Source: Authors' calculations (2023).

**Table 7. Forecast-error variance decompositions of food insecurity.**

| | FIMI | | WUEA | | WUEI | | WUIS | |
|---|---|---|---|---|---|---|---|---|
| Step | oirf | fevd | oirf | fevd | oirf | fevd | oirf | fevd |
| 1 | 103.311 | 1 | 05.175 | 96.9237 | 28.7812 | 60.2911 | 5.01372 | 38.9605 |
| 2 | 93.7057 | 99.2984 | 04.341 | 96.0716 | 27.5491 | 61.289 | 4.72672 | 40.8502 |
| 3 | 85.3492 | 97.7956 | 03.5027 | 93.8656 | 26.5258 | 60.9774 | 4.46336 | 42.0978 |
| 4 | 77.9414 | 95.6034 | 02.7907 | 90.6413 | 25.5339 | 60.0773 | 4.2086 | 42.548 |
| 5 | 70.7667 | 92.9139 | 02.2119 | 86.9066 | 24.6684 | .8.5922 | 3.97755 | 42.2499 |
| 6 | 64.2496 | 89.8743 | 01.7403 | 82.8999 | 23.8504 | 56.9068 | 3.75752 | 41.6636 |
| 7 | 58.5895 | 86.7232 | 01.371 | 78.7141 | 23.211 | 55.109 | 3.53356 | 40.7014 |
| 8 | 53.5117 | 83.4074 | 01.0942 | 74.7004 | 22.4683 | 53.1881 | 3.30132 | 39.6086 |
| 9 | 48.7207 | 80.3023 | 00.8745 | 70.939 | 21.9109 | 51.1579 | 3.09893 | 38.3812 |
| 10 | 44.6465 | 77.417 | 00.6939 | 67.4039 | 21.1962 | 49.4933 | 2.91043 | 36.8322 |

Posterior medians reported.

95% HPD credible lower and upper bounds reported.

(1) irfname = birf, impulse = FIMI1, WUEA, WUEI and WUIS and response = owned variables.

(2) Oirf = Orthogonal IRFs (OIRFs) explain the impulse response to a one-standard-deviation shock.

Source: Author analysis, 2023.

correlation was observed. There are stability conditions with no autocorrelation in lag order and it was decided that in the short-run, changes in water use efficiency have an explanatory power concerning the current FIMI. The coefficients of the water use efficiency are significant on the FIMI in the short run, meaning that food insecurity is better explained by water use efficiency management for various purposes.

Moreover, this study concludes that water use efficiency management for agriculture and industrial purposes reduces food insecurity while increasing the water available for service use increases food insecurity. The results consistently suggest that the shocks of water management for agricultural use have a significant and continual impact on food insecurity. The shocks in food insecurity could be considered self-shocks during the last 10 years. In contrast, the water use efficiency for agricultural purposes appears to be the highest among the other two uses when explaining the variance in food insecurity. Across the 10-year period, the water uses efficiency shocks decrease food insecurity. To conclude, the novel findings are that water management shocks affect food insecurity in Saudi Arabia. The future evaluation of food security and other development-related sectors such as inflation, population, energy, prices, and climate change factors, as they interact, will determine the future reduction of food insecurity in Saudi Arabia. Ultimately, this study provides a strong recommendation directed at decision-makers that may support food security in Saudi Arabia through implementing and enhancing water use efficiency management as an integral part of achieving the SDGs in Saudi Arabia.

## Supporting information

**S1 File.**
(XLSX)

## Author Contributions

**Conceptualization:** Raga M. Elzaki.

**Data curation:** Raga M. Elzaki.

**Formal analysis:** Raga M. Elzaki.

**Funding acquisition:** Raga M. Elzaki.

**Investigation:** Raga M. Elzaki, Mohammed Al-Mahish.

**Methodology:** Raga M. Elzaki, Mohammed Al-Mahish.

**Software:** Mohammed Al-Mahish.

**Writing – original draft:** Raga M. Elzaki.

**Writing – review & editing:** Raga M. Elzaki, Mohammed Al-Mahish.

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
