## [Decision Letter · Decision Letter 0]

6 Sep 2023

PONE-D-23-19746Food Insecurity and Water Management Shocks in Saudi Arabia: Bayesian VAR AnalysisPLOS ONE

Dear Dr. Elzaki,

Thank you for submitting your manuscript to PLOS ONE. After careful consideration, we feel that it has merit but does not fully meet PLOS ONE’s publication criteria as it currently stands. Therefore, we invite you to submit a revised version of the manuscript that addresses the points raised during the review process.

We look forward to receiving your revised manuscript.

Kind regards,

Shujahat Haider Hashmi, PhD Regional Economics

Academic Editor

PLOS ONE

Journal Requirements:

"This research was funded by the Deanship of Scientific Research, King Faisal University, Al-Ahsa, Saudi Arabia for financial support under the Ambitious Researcher Track."

"NO authors have competing interests."

Reviewers' comments:

Reviewer's Responses to Questions

**Comments to the Author**

1. Is the manuscript technically sound, and do the data support the conclusions?

Reviewer #1: Partly

Reviewer #2: Yes

2. Has the statistical analysis been performed appropriately and rigorously? 

Reviewer #1: No

Reviewer #2: Yes

3. Have the authors made all data underlying the findings in their manuscript fully available?

Reviewer #1: No

Reviewer #2: Yes

4. Is the manuscript presented in an intelligible fashion and written in standard English?

Reviewer #1: No

Reviewer #2: No

5. Review Comments to the Author

Reviewer #1: Introduction

-First paragraph- spelling- macr-level- should be Macro-

- second paragraph- " exchange rates(EXR)- The exchange rate is the value of one currency relative to another currency. In the context you provided, it is unclear which currency or currencies are being referred to.

- After Figure 1 - Note: explained equations - I’m not sure what these equations mean? The authors need to explain with a foot note if possible

-"The contribution of this paper.. " need to explore in detailed - need to explain "why the paper deserve to publish?' what is the contribution to knowledge - need to explain in detailed with the alignment of literature- what is done ( by referring recent studies up to 12)- then find the gap- explain - the contribution of the knowledge or science explicitly

-The introduction provides overview of the topic of food insecurity and its various factors, as well as the need for research in the context of Saudi Arabia. However, there are a few recommendations to improve the clarity and structure of the introduction: Clear research objective: While the introduction mentions the aim of identifying and estimating the Food Insecurity Multidimensional Index (FIMI) and examining the dynamic effect of sustainable water on food insecurity, it would be helpful to explicitly state the research objective in a clear and concise manner. For example, you could specify whether the objective is to assess the impact of water efficiency on food insecurity in Saudi Arabia or to analyze the factors influencing FIMI in the Middle East.

Date and Models

-It will be useful to mention which statistical software was used to conduct the analysis in

- page 8- The article does not explicitly mention whether the author considered the possibility of multiple structural breaks in their data. If the author did not consider the possibility of multiple structural breaks in the variables, it should be mentioned as a limitation of the study. The absence of testing for multiple structural breaks could be considered a limitation of the study. Ignoring the possibility of multiple breaks may lead to biased parameter estimates. Clemente-montanes-reyes unit root test is a test capable of testing for two structural breaks (for future reference).

-Page 9- equation 5-6- Just keep in mind that the three models test for the source of non-stationarity. For example if the source of non-stationary is due to a deterministic trend, then one should detrend the data and not difference the data. Differencing a deterministic trend does not make the data stationary but instead introduces a unit root, which makes the problem even worse. To address a deterministic trend, other techniques can be employed, such as detrending the data using regression analysis. Therefore on should be diligent in identifying the source of non-stationarity in the data since it will determine which transformation will be applied to address non-stationarity. Based on the aforementioned I would advise the authors to make sure that model C is the correct model to be applied.

Results and discussion

-page 25- (Forecast-error variance decompositions)- This paper do not have a discussion therefore I expect to see in the results somewhere references included that interact with the study’s results. The authors should for example see if they can find studies that have illustrated/found similarly that water use efficiency management for agriculture and industrial purposes reduces food insecurity while increasing water for services use increases food insecurity.

Reviewer #2: The authors have done a good analysis but the paper is not organized appropriately. English usage in the paper needs to be improved as the paper many sections / sentences in the paper are difficult to be read and understood. The conclusions of the paper are not properly drawn and presented based on the analysis. The conclusion is mainly repetitions of what is explained in previous sections.

The abstract needs to be rewritten to highlight the major findings based on the analysis done.

The paper includes many Tables and Figures. The authors may try to combine those or reduce the number of Tables / Figures.

6. PLOS authors have the option to publish the peer review history of their article (what does this mean?). If published, this will include your full peer review and any attached files.

Reviewer #1: **Yes: **Yonas T. Bahta

Reviewer #2: No

---

## [Author Response · Author response to Decision Letter 0]

6 Oct 2023

PONE-D-23-19746: Food Insecurity and Water Management Shocks in Saudi Arabia: Bayesian VAR Analysis

Dear Editor Journal of PLOS One.

First thanks for all the suggestions and spotting points. Our answers with details in red color after each suggestion in this file. In the main text, we responded to all suggestions with track and red color.

Reviewer #1: Introduction

-First paragraph- spelling- macr-level- should be Macro-

Answer: Corrected to Macro in line 133.

- second paragraph- " exchange rates(EXR)- The exchange rate is the value of one currency relative to another currency. In the context you provided, it is unclear which currency or currencies are being referred to.

Answer: the exchange rate is clarified in line 57 also we insert the footnote for more information. 

- After Figure 1 - Note: explained equations - I’m not sure what these equations mean? The authors need to explain with a foot note if possible.

Answer: The equations were explained as Foote notes in lines 66-67.

-"The contribution of this paper. " need to explore in detailed - need to explain "why the paper deserve to publish?' what is the contribution to knowledge - need to explain in detailed with the alignment of literature- what is done ( by referring recent studies up to 12)- then find the gap- explain - the contribution of the knowledge or science explicitly.

Answer: The contribution of the paper was explored in detail with the alignment of literature in lines 89-125.

-The introduction provides overview of the topic of food insecurity and its various factors, as well as the need for research in the context of Saudi Arabia. However, there are a few recommendations to improve the clarity and structure of the introduction: Clear research objective: While the introduction mentions the aim of identifying and estimating the Food Insecurity Multidimensional Index (FIMI) and examining the dynamic effect of sustainable water on food insecurity, it would be helpful to explicitly state the research objective in a clear and concise manner. For example, you could specify whether the objective is to assess the impact of water efficiency on food insecurity in Saudi Arabia or to analyze the factors influencing FIMI in the Middle East.

Answer: We added the objective of the study: to assess the impact of water efficiency on food insecurity in Saudi Arabia in lines 85-86.

Date and Models

-It will be useful to mention which statistical software was used to conduct the analysis in

- page 8- The article does not explicitly mention whether the author considered the possibility of multiple structural breaks in their data. If the author did not consider the possibility of multiple structural breaks in the variables, it should be mentioned as a limitation of the study. The absence of testing for multiple structural breaks could be considered a limitation of the study. Ignoring the possibility of multiple breaks may lead to biased parameter estimates. Clemente-montanes-reyes unit root test is a test capable of testing for two structural breaks (for future reference).

It will be useful to mention which statistical software was used to conduct the analysis in

- page 8-

Answer: Statistical software was mentioned in line 210.

The absence of testing for multiple structural breaks could be considered a limitation of the study.

Answer: We analyzed the multiple structural breaks and the results are included in Table (2) in red color. 

-Page 9- Equations 5-6- Just keep in mind that the three models test for the source of non-stationarity. For example if the source of non-stationary is due to a deterministic trend, then one should detrend the data and not difference the data. Differencing a deterministic trend does not make the data stationary but instead introduces a unit root, which makes the problem even worse. To address a deterministic trend, other techniques can be employed, such as detrending the data using regression analysis. Therefore, on should be diligent in identifying the source of non-stationarity in the data since it will determine which transformation will be applied to address non-stationarity. Based on the aforementioned I would advise the authors to make sure that model C is the correct model to be applied.

Answer: Yes, you’re perfect. Trend-stationarity would imply no unit root for the detrended series. Model C has solved these mentioned problems. Because it depends on also on constants. And for that, we accept the VAR and Bayesian VAR models. Equation (8) in line 281. Also, Bayesian VAR models are more flexible than traditional VAR and can handle both stationary and non-stationary data. 

Results and discussion

-page 25- (Forecast-error variance decompositions)- This paper do not have a discussion therefore I expect to see in the results somewhere references included that interact with the study’s results. The authors should for example see if they can find studies that have illustrated/found similarly that water use efficiency management for agriculture and industrial purposes reduces food insecurity while increasing water for services use increases food insecurity.

Answer: References for confirmation were added in lines 499-502.

Reviewer #2: The authors have done a good analysis but the paper is not organized appropriately. 

Answer: We organized the paper.

English usage in the paper needs to be improved as the paper many sections / sentences in the paper are difficult to be read and understood.

Answer: The article's English usage was revised carefully with a track. 

 The conclusions of the paper are not properly drawn and presented based on the analysis. The conclusion is mainly repetitions of what is explained in previous sections.

Answer: We rewritten the conclusion section.

The abstract needs to be rewritten to highlight the major findings based on the analysis done.

Answer: The abstract was rewritten. 

The paper includes many Tables and Figures. The authors may try to combine those or reduce the number of Tables / Figures.

 Answer: We combined tables 2 and 3 in table 2 (in lines 402-403), and tables 5 and 6 in table 4(in lines 419). Also figure 3, 4, and 5 are combined in one figure named 3A,3B, and 3C in lines 453).

---

## [Decision Letter · Decision Letter 1]

7 Nov 2023

PONE-D-23-19746R1Food Insecurity and Water Management Shocks in Saudi Arabia: Bayesian VAR AnalysisPLOS ONE

Dear Dr. Elzaki,

Thank you for submitting your manuscript to PLOS ONE. After careful consideration, we feel that it has merit but does not fully meet PLOS ONE’s publication criteria as it currently stands. Therefore, we invite you to submit a revised version of the manuscript that addresses the points raised during the review process.

We look forward to receiving your revised manuscript.

Kind regards,

Shujahat Haider Hashmi, PhD Regional Economics

Academic Editor

PLOS ONE

**Additional Editor Comments:**

As you can see, the reviewers have thoroughly evaluated your revised version, and they have appreciated your efforts. However, there are still some critical points that need to be addressed before the possible acceptance of the manuscript to further improve the quality of the manuscript. Therefore, you are requested to make revisions as per the given comments and submit the revised version again for further assessment.

Reviewers' comments:

Reviewer's Responses to Questions

**Comments to the Author**

1. If the authors have adequately addressed your comments raised in a previous round of review and you feel that this manuscript is now acceptable for publication, you may indicate that here to bypass the “Comments to the Author” section, enter your conflict of interest statement in the “Confidential to Editor” section, and submit your "Accept" recommendation.

Reviewer #2: (No Response)

Reviewer #3: (No Response)

2. Is the manuscript technically sound, and do the data support the conclusions?

Reviewer #2: Yes

Reviewer #3: Partly

3. Has the statistical analysis been performed appropriately and rigorously? 

Reviewer #2: Yes

Reviewer #3: I Don't Know

4. Have the authors made all data underlying the findings in their manuscript fully available?

Reviewer #2: Yes

Reviewer #3: Yes

5. Is the manuscript presented in an intelligible fashion and written in standard English?

Reviewer #2: Yes

Reviewer #3: Yes

6. Review Comments to the Author

Reviewer #2: The authors have taken utmost efforts in improving the manuscript. It is improved a lot. However, you have to improve English further, prior to publish it.

Reviewer #3: 1. The manuscript titled “Food Insecurity and Water Management Shocks in 1 Saudi Arabia: Bayesian VAR Analysis” aims to evaluate the influence of water use efficiency as a Sustainable Development Goal on the Food Insecurity Multidimensional Index. Additionally, it estimates the impact of water efficiency on food insecurity in Saudi Arabia using the Bayesian Vector Autoregressive model. The study holds promise as a valuable contribution, but in its present form, the manuscript is too lengthy and appears to have a predominantly local focus while lacking a broader global/scientific appeal. Nonetheless, I have the following suggestions to the authors, which can enhance the manuscript:

2. The are so many typos and grammatical error in the manuscript. Please revisit.

3. The study is very regional. Although it will be helpful for local communities, how useful is it to the international water community?

4. Abstract: The abstract should articulate and highlight the research problem in a broader perspective so that it is easy for the readers to easily connect to the article. The objectives should be clearly mentioned. The approach and the data used, along with the findings and their implications can be articulated subsequently. Therefore, re-writing is required in this regard. Also try to be more concise in writing without losing the bigger picture.

5. Introduction: The introduction could benefit from a clearer statement of the research problem identified based on a more comprehensive review of the recent literature, articulation of testable hypotheses, and objectives.

6. Support your sentence in the introduction section with proper references such as https://doi.org/10.1016/j.ecolind.2023.110766;
https://doi.org/10.3390/w15132438;
https://doi.org/10.1007/s13201-023-01928-z.

7. It is better to include a flowchart of the methodology.

8. The resolution of all the figures should be increased.

9. There are so many abbreviations used in the manuscript. It is suggested to insert the table of abbreviations.

10. Please check that the reference should be given to all the equations used in the manuscript.

11. Results: Please try to limit the number of tables as it will only confuse the reader. Only the most important ones can be kept in the manuscript, while additional/supporting tables and figures may be given as supplementary materials. Please maintain conciseness without neglecting the important findings. The references also need to be updated.

12. Conclusions: This section needs re-writing and should effectively synthesize the essential points presented in the preceding parts and make a succinct statement that reiterates the text's central message or core theme. It should leave the reader with a lasting impression and bring together the topic's overall relevance or ramifications.

13. Overall, the manuscript's general readability is hampered by a number of unclear/clumsily constructed sentences. Improving the clarity of language and sentence structure throughout the manuscript would enhance the readability and overall quality of the presentation.

7. PLOS authors have the option to publish the peer review history of their article (what does this mean?). If published, this will include your full peer review and any attached files.

Reviewer #2: No

Reviewer #3: No

---

## [Author Response · Author response to Decision Letter 1]

19 Nov 2023

Response to the reviewer 3 comments for article: Food Insecurity and Water Management Shocks in Saudi Arabia: Bayesian VAR Analysis

Thank you very much for your efforts and taking the time to review this manuscript. Thanks for all the suggestions and spotting points. We responded to all suggestions. Please find the detailed responses below and the corresponding revisions/corrections highlighted/in track changes and red color in the re-submitted revised (2) files. 

Reviewer #3: 1. The manuscript titled “Food Insecurity and Water Management Shocks in 1 Saudi Arabia: Bayesian VAR Analysis” aims to evaluate the influence of water use efficiency as a Sustainable Development Goal on the Food Insecurity Multidimensional Index. Additionally, it estimates the impact of water efficiency on food insecurity in Saudi Arabia using the Bayesian Vector Autoregressive model. The study holds promise as a valuable contribution, but in its present form, the manuscript is too lengthy and appears to have a predominantly local focus while lacking a broader global/scientific appeal. Nonetheless, I have the following suggestions to the authors, which can enhance the manuscript:

2. The are so many typos and grammatical error in the manuscript. Please revisit.

Answer: Thanks for pointing out this. We revised the grammar errors.

3. The study is very regional. Although it will be helpful for local communities, how useful is it to the international water community?

Response: Thanks for pointing out this comment. We briefly explain the benefit of this study to the global community along the 137-142.

4. Abstract: The abstract should articulate and highlight the research problem in a broader perspective so that it is easy for the readers to easily connect to the article. The objectives should be clearly mentioned. The approach and the data used, along with the findings and their implications can be articulated subsequently. Therefore, re-writing is required in this regard. Also try to be more concise in writing without losing the bigger picture.

Response: Thank you for mentioning this. The abstract was rewritten accordingly. 

5. Introduction: The introduction could benefit from a clearer statement of the research problem identified based on a more comprehensive review of the recent literature, articulation of testable hypotheses, and objectives.

Response: Thanks for mentioning this. We cited the references mentioned below to enrich the introduction. 

6. Support your sentence in the introduction section with proper references such as https://doi.org/10.1016/j.ecolind.2023.110766;
https://doi.org/10.3390/w15132438;
https://doi.org/10.1007/s13201-023-01928-z.

Response: So thank for suggesting these three references. We cited all of them.

Comment: https://doi.org/10.1016/j.ecolind.2023.110766: 

Response: Very thankful for suggesting this reference. We cited as reference no. 14

Comment: https://doi.org/10.3390/w15132438: 

Response: Very thankful for suggesting this reference. We cited as reference no. 13

Comment: https://doi.org/10.1007/s13201-023-01928-z: 

Response: Very thankful for suggesting this reference. We cited as reference no. 15

7. It is better to include a flowchart of the methodology.

Response: thanks for suggesting this. A flowchart was created in the methodology section in lines: 223-234. 

8. The resolution of all the figures should be increased.

Response: Thanks for suggesting this. All the figure's resolutions were increased. 

9. There are so many abbreviations used in the manuscript. It is suggested to insert the table of abbreviations.

Response: thanks for your comment. We already inserted the abbreviations in lines 546-575.

10. Please check that the reference should be given to all the equations used in the manuscript.

Response: All references were given for all equations. 

11. Results: Please try to limit the number of tables as it will only confuse the reader. Only the most important ones can be kept in the manuscript, while additional/supporting tables and figures may be given as supplementary materials. Please maintain conciseness without neglecting the important findings. The references also need to be updated.

Response: Thanks for this comment. We reduced the tables, and we kept the important ones.

The references also need to be updated.

Response: We updated some references, e.g., 91-96, but some references about the equations (initial investors of equations) we can’t update them.

12. Conclusions: This section needs re-writing and should effectively synthesize the essential points presented in the preceding parts and make a succinct statement that reiterates the text's central message or core theme. It should leave the reader with a lasting impression and bring together the topic's overall relevance or ramifications.

Response: thanks for pointing out these points. We rewrite the conclusions accordingly.

13. Overall, the manuscript's general readability is hampered by a number of unclear/clumsily constructed sentences. Improving the clarity of language and sentence structure throughout the manuscript would enhance the readability and overall quality of the presentation.

Response: Thanks for your comment. We rechecked the manuscript accordingly.

---

## [Decision Letter · Decision Letter 2]

18 Dec 2023

Food Insecurity and Water Management Shocks in Saudi Arabia: Bayesian VAR Analysis

PONE-D-23-19746R2

Dear Dr. Elzaki,

We’re pleased to inform you that your manuscript has been judged scientifically suitable for publication and will be formally accepted for publication once it meets all outstanding technical requirements.

Kind regards,

Shujahat Haider Hashmi, PhD Regional Economics

Academic Editor

PLOS ONE

Additional Editor Comments (optional):

It is to inform that the reviewers have now accepted your revisions and they recommend acceptance of your paper. Therefore, we wish you best of luck for your future publication.

Reviewers' comments:

Reviewer's Responses to Questions

**Comments to the Author**

1. If the authors have adequately addressed your comments raised in a previous round of review and you feel that this manuscript is now acceptable for publication, you may indicate that here to bypass the “Comments to the Author” section, enter your conflict of interest statement in the “Confidential to Editor” section, and submit your "Accept" recommendation.

Reviewer #2: All comments have been addressed

Reviewer #3: All comments have been addressed

2. Is the manuscript technically sound, and do the data support the conclusions?

Reviewer #2: Yes

Reviewer #3: Yes

3. Has the statistical analysis been performed appropriately and rigorously? 

Reviewer #2: Yes

Reviewer #3: Yes

4. Have the authors made all data underlying the findings in their manuscript fully available?

Reviewer #2: Yes

Reviewer #3: Yes

5. Is the manuscript presented in an intelligible fashion and written in standard English?

Reviewer #2: Yes

Reviewer #3: Yes

6. Review Comments to the Author

Reviewer #2: (No Response)

Reviewer #3: In view of mine, revised version of manuscript is suitable for publication in present form, all the best

7. PLOS authors have the option to publish the peer review history of their article (what does this mean?). If published, this will include your full peer review and any attached files.

Reviewer #2: No

Reviewer #3: No

---

## [Editor Report · Acceptance letter]

11 Jan 2024

PONE-D-23-19746R2 

PLOS ONE

Dear Dr. Elzaki, 

I'm pleased to inform you that your manuscript has been deemed suitable for publication in PLOS ONE. Congratulations! Your manuscript is now being handed over to our production team.

Kind regards, 

on behalf of

Dr. Shujahat Haider Hashmi 

Academic Editor

PLOS ONE